Everything you always wanted to know about gene flow in tropical landscapes (but were afraid to ask)

Monteiro Waléria Pereira 1
Veiga Jamille Costa 2
Silva Amanda Reis 3
Carvalho Carolina da Silva 1
Lanes Éder Cristian Malta 1
Rico Yessica 4
Jaffé Rodolfo r.jaffe@ib.usp.br rodolfo.jaffe@itv.org 1 5
1 Instituto Tecnológico Vale , Belém , PA , Brazil
2 Instituto de Ciências Biológicas, Universidade Federal do Pará , Belém , Pará , Brazil
3 Departamento de Botânica, Museu Paraense Emílio Goeldi , Belém , Pará , Brazil
4 CONACYT, Red de Diversidad Biológica del Occidente Mexicano, Instituto de Ecología, A.C. , Michoacán , Mexico
5 Departamento de Ecologia, Universidade de São Paulo , São Paulo , Brazil
Giordani Paolo
Electronic publication date: 2019 Feb 13
Publication date: 2019
Volume: 7
Electronic Location ID: e6446
Received 2018 Aug 28; Accepted 2019 Jan 15
Copyright: ©2019 Monteiro et al.
Copyright year: 2019
Copyright holder: Monteiro et al.
License: This is an open access article distributed under the terms of the Creative Commons Attribution License, which permits unrestricted use, distribution, reproduction and adaptation in any medium and for any purpose provided that it is properly attributed. For attribution, the original author(s), title, publication source (PeerJ) and either DOI or URL of the article must be cited.
License URL: https://creativecommons.org/licenses/by/4.0/

Keywords: Functional connectivity, Landscape genetics, Isolation by resistance, Matrix permeability, Tropical biodiversity

Funding: Instituto Tecnologico Vale, CAPES 88882.161651/2017-01 15/2014 88887.156652/2017-00 CNPq 300714/2017-3 131105/2016-7 301616/2017-5 This study was funded by Instituto Tecnologico Vale, CAPES (88882.161651/2017-01–Waléria Pereira Monteiro, 15/2014–Jamille Costa Veiga and 88887.156652/2017-00–Carolina da Silva Carvalho), and CNPq (300714/2017-3–Éder Cristian Malta Lanes, 131105/2016-7–Amanda Reis Silva and 301616/2017-5–Rodolfo Jaffé). The funders had no role in study design, data collection and analysis, decision to publish, or preparation of the manuscript.

==============================
The bulk of the world’s biodiversity is found in tropical regions, which are increasingly threatened by the human-led degradation of natural habitats. Yet, little is known about tropical biodiversity responses to habitat loss and fragmentation. Here we review all available literature assessing landscape effects on gene flow in tropical species, aiming to help unravel the factors underpinning functional connectivity in the tropics. We map and classify studies by focus species, the molecular markers employed, statistical approaches to assess landscape effects on gene flow, and the evaluated landscape and environmental variables. We then compare qualitatively and quantitatively landscape effects on gene flow across species and units of analysis. We found 69 articles assessing landscape effects on gene flow in tropical organisms, most of which were published in the last five years, were concentrated in the Americas, and focused on amphibians or mammals. Most studies employed population-level approaches, microsatellites were the preferred type of markers, and Mantel and partial Mantel tests the most common statistical approaches used. While elevation, land cover and forest cover were the most common gene flow predictors assessed, habitat suitability was found to be a common predictor of gene flow. A third of all surveyed studies explicitly assessed the effect of habitat degradation, but only 14 of these detected a reduced gene flow with increasing habitat loss. Elevation was responsible for most significant microsatellite-based isolation by resistance effects and a single study reported significant isolation by non-forested areas in an ant. Our study reveals important knowledge gaps on the study of landscape effects on gene flow in tropical organisms, and provides useful guidelines on how to fill them.

Introduction

About two-thirds of all known species occur in tropical forests and the majority of the world’s most threatened biodiversity hotspots are in the tropics (Myers et al., 2000; Brown, 2014). Extinction rates from habitat loss and fragmentation are acute in the region, and the degradation of essential ecosystem functions and services are threatening billions of people living in tropical countries (Bradshaw, Sodhi & Brook, 2009). However, the vast majority of studies assessing biodiversity responses to habitat degradation have been undertaken in temperate regions due to a lower investment in research and development in tropical countries (Collen et al., 2008; Barlow et al., 2018). For instance, a recent analysis of 182 studies describing links between biodiversity and ecosystem function (Clarke et al., 2017) found that only 13% were carried in the tropics, and nearly half of these (42%) were conducted in a single country (Costa Rica). There is thus a pressing need to reduce the knowledge gap concerning the impact of the degradation of natural habitats on tropical biodiversity.

Community-level approaches assessing biodiversity responses to habitat degradation have focused on measuring changes in species richness, composition, and the abundance of indicator species (Morin, 2009). While these metrics underpin ecosystem function, they may not always be the best proxies to detect rapid responses to habitat loss and fragmentation. Local species extinctions may occur after long periods of time since the onset of disturbance (Jackson & Sax, 2010), whereas species abundance can be affected by multiple environmental or demographic factors unrelated to habitat degradation (Ehrlén & Morris, 2015). Moreover, complex inter-specific interactions can make natural communities resilient to environmental change and thus mask the effect of habitat degradation on community composition (Devictor, Julliard & Jiguet, 2008). Instead, population-level metrics based on genetic information can offer a higher resolution to detect rapid responses to environmental change (Manel & Holderegger, 2013). For instance, changes in genetic diversity and gene flow patterns in response to recent landscape modification have been found across several species (Balkenhol et al., 2016; DiLeo & Wagner, 2016), although tropical organisms have been rarely assessed (Storfer et al., 2010).

Even though the effects of habitat loss and fragmentation on genetic diversity have been reviewed extensively (Aguilar et al., 2006; Aguilar et al., 2008; Keyghobadi, 2007; Vranckx et al., 2012; Lino et al., in press; Schlaepfer et al., 2018), there is an important knowledge gap regarding general landscape effects on gene flow (DiLeo & Wagner, 2016). By influencing the willingness of an organism to cross a particular environment and the physiological or fitness costs of moving through it, the resistance imposed by landscape structure on the dispersal of organisms can ultimately affect genetic differentiation and patterns of gene flow (Zeller, McGarigal & Whiteley, 2012; Balkenhol et al., 2016). To understand which landscape features impose a greater resistance on gene flow, landscape geneticists first create resistance surfaces for landscape variables of interest, then use these surfaces to estimate cost or resistance distances between sampling locations, and finally regress measures of gene flow on these resistance distances (Spear, Cushman & McRae, 2016). Significant associations between gene flow metrics and landscape resistance distances are taken for evidence of isolation by resistance (IBR), and effect sizes can be considered proxies of functional connectivity (Manel & Holderegger, 2013).

Understanding the factors underpinning functional connectivity across species is essential to design ecological corridors, identify conservation units, assess population threat status, optimize pathogen and invasive species management, assist planning of natural heritage systems, and restore threatened populations (Bowman et al., 2016; Waits, Cushman & Spear, 2016). However, no efforts have yet been made to gather, standardize and compare IBR effects across studies and organisms. For instance, landscape genetics is still a young field of research (Manel & Holderegger, 2013), and the vast majority of landscape genetic studies have focused in a single species (DiLeo & Wagner, 2016; Waits, Cushman & Spear, 2016). So far, gene flow has been shown to be influenced by various factors, including forest cover, land cover, topography, roads, rivers, and climate, but responses vary greatly across species and units of analysis (populations or individuals; see Balkenhol et al., 2016 and references therein).

Aiming to unravel the main drivers of functional connectivity in tropical landscapes, here we compiled all studies that assessed landscape effects on gene flow in tropical species so far. To our knowledge, this work represents the first quantitative comparison of such effects across species and units of analysis. We believe this systematic review can help characterize the current knowledge gap on tropical biodiversity responses to habitat degradation, and thereby highlight future research needs.

Survey methodology

Dataset

We employed the following search engines to perform a recursive literature search of landscape effects on gene flow in tropical species published by June 2018: Scielo (http://www.scielo.org), Portal de Periódicos da Coordenação de Aperfeiçoamento de Pessoal de Nível Superior do Ministério da Educação (CAPES/MEC) (http://www.periodicos.capes.gov.br/); Google Scholar (https://scholar.google.com.br/); Web of Knowledge (http://www.isiknowledge.com), and Scopus (http://www.scopus.com). We used the following combination of keywords and Boolean operators: (“landscape resistance” or landscape or resistance or fragmentation or “land use” or “habitat loss” or deforestation) and (genetic* or “genetic differentiation” or “gene flow” or “genetic distance” or FST or relatedness or kinship). Articles containing at least one of the keywords on each side of the “and” operator were analyzed along with the relevant references therein. Even though this search approach may not be easily replicated (as it involves a substantial effort), it is more likely to minimize omissions than approaches based on the results obtained from search engines alone. We then identified those studies that explicitly related landscape with gene flow metrics in organisms collected between the tropics of Cancer and Capricorn (23.5° north and south of the equator) or within 200 km from them. Articles addressing only isolation by geographic distance (IBD) were excluded, as our aim was to survey studies that specifically incorporated landscape effects on gene flow in addition to geographic distance. We then gathered all available information on the study objectives, focus species, study site, ecosystem, the extent of the study area, the unit of analysis employed, sample size, types and number of genetic markers employed, genetic and landscape resistance metrics employed, statistical methods, landscape or environmental predictors assessed, and the effects reported.

Comparing landscape effects on gene flow across studies

We performed both qualitative and quantitative comparisons of landscape effects on gene flow across studies. For the former, we grouped studies by the landscape or environmental factors assessed and the focus taxonomic group, and summarized the reported effects on gene flow. For the quantitative comparison we selected a subset of our dataset containing only studies that: (i) explicitly reported correlation or regression coefficients, calculated from at least three samples, and (ii) employed nuclear microsatellite markers to measure gene flow, given that measures of genetic differentiation obtained with different genetic markers are not directly comparable across studies (Wan et al., 2004; Allendorf, Luikart & Aitken, 2013). We then separated the studies fulfilling these requirements in two groups according to the units of analysis employed: Those using population-level metrics of genetic differentiation (FST or Dest), and those using individual-level metrics of genetic distance (Rousset’s a, relatedness and kinship; Dataset S3). We note that these genetic distance metrics should be considered surrogates of actual gene flow, as they reflect the joint influence of genetic drift and dispersal (Prunier et al., 2017). An effect size approach was used to compare isolation by resistance (IBR) within both types of studies (individual and population-level). Correlation coefficients were first normalized using the Fisher’s z-transformation (z), and standard errors (se) were calculated as following: z=12ln1+r1−r

se=1N−3.

Where r is the correlation coefficient, ln the natural logarithm and N the number of pairwise comparisons (between individuals or populations). Effect sizes were then calculated dividing the normalized correlation coefficients and standard errors (z/se) (Ellis, 2010). Effect sizes of regression coefficients were calculated dividing them by their respective standard errors. To facilitate comparisons between population-level and individual-level metrics of genetic differentiation we inverted the sign of relatedness and kinship estimates, thus representing genetic dissimilarity. We also calculated 95% confidence intervals for all effect sizes and retrieved the statistical significance (p-values) of IBR effects reported in the original studies. We note that effect sizes were only used for comparative purposes, and that as in previous reviews (DiLeo & Wagner, 2016), small sample sizes did not allow performing a formal meta-analysis.

Results

We found a total of 69 articles assessing landscape effects on gene flow in tropical organisms (Dataset S1), most of which were undertaken in the Americas (Fig. 1). We recorded 154 target species belonging to eight major taxonomic groups, from which amphibians contained the largest number of species and mammals the highest number of papers (Fig. 2). Most focus species were terrestrial and only three exclusively aquatic species were evaluated. The majority of studies analyzed a single species, but nine publications evaluated two or more. Three studies contributed with more than 40% of all recorded species (Wang, Glor & Losos, 2013; Paz et al., 2015; Jaffé et al., 2016). The oldest study found in our literature search (Trénel et al., 2008) investigated the impact of contemporary Andean landscape features on the spatial genetic structure of a palm tree. After this work, we observed a jump in the number of publications from 2013 onwards (Fig. 3).

Figure 1 Sampling locations of the surveyed studies.

Taxonomic groups are indicated by colors and the unit of analysis by shapes (triangles indicate individual-level studies and circles population-level ones). Horizontal dotted lines represent the Tropic of Cancer, the Equator and the Tropic of Capricorn respectively, from North to South.

Figure 2 Number of species assessed and number of publications for each taxonomic group.

Figure 3 Number of studies assessing landscape effects on gene flow in tropical organisms, published between 2008 and 2018.

The surveyed studies often had overlapping objectives, which comprised assessing contemporary and historical effects of climate on gene flow (Trénel et al., 2008; Ramírez-Barahona & Eguiarte, 2014); predicting gene flow with habitat suitability models (Poelchau & Hamrick, 2012; Guarnizo & Cannatella, 2013; Paz et al., 2015); assessing landscape and climatic effects on gene flow (Hohnen et al., 2016; Lanes et al., 2018); identifying dispersal routes (Andraca-Gómez et al., 2015; Cleary, Waits & Finegan, 2017; Thatte et al., 2018) and barriers to gene flow (Robertson, Duryea & Zamudio, 2009; Boff et al., 2014; Oliveira et al., 2017); and evaluating the impact of habitat fragmentation on gene flow (Balkenhol et al., 2013; Joshi et al., 2013; De Campos Telles et al., 2014; Carvalho et al., 2015; Ruiz-Lopez et al., 2015).

Five types of molecular markers were found across all studies (Fig. 4), and only five publications used more than one type of marker (usually microsatellites and mtDNA). Microsatellites were the most frequently used markers, with more studies using them than publications using all other markers combined. More than 70% of all studies were performed at the population-level and only five studies used both population and individual-level approaches (Dataset S1). Electrical resistance (ER) was the most common resistance metric employed (McRae, 2006), and Mantel and partial Mantel test the most common statistical methods used to relate genetic with resistance distances (Fig. 5).

Figure 4 Proportion of studies using different types of genetic markers to assess landscape effects on gene flow in tropical organisms.

Microsatellites, AFLPs and SNPs refer to nuclear DNA. Mitochondrial DNA (mtDNA) and Chloroplast DNA (cpDNA) are specified as such.

Figure 5 Number of studies using different statistical approaches to assess landscape effects on gene flow in tropical organisms.

Full methods names, by order of appearance on the figure are: Mantel and Partial Mantel tests, Maximum likelihood population effects (MLPE) models, generalized dissimilarity models (GDM), redundancy analyses (RDA), multiple regression on distance matrices (MRDM), generalized linear models (GLM), linear mixed-effect models (LMM), Monte Carlo permutation matrix regression technique (MCPMRT), matrix regression approach (MRA), Random Forest Analysis (RFA), Structural equation modelling (SEM), and Linear Model with Permutation (LMP).

Landscape and environmental predictors of gene flow included elevation (altitude, terrain ruggedness and slope), land cover, forest cover, habitat suitability (derived from species distribution models), water (rivers, streams and the ocean), precipitation, roads and temperature (Fig. 6, Dataset S1). Only six out of 22 studies considering elevation, four out of 22 studies evaluating land cover, four out of 21 studies assessing forest cover, and 14 out of 17 studies relying on habitat suitability models reported an effect on gene flow (Table S1). Most plant and amphibian studies used habitat suitability models to generate resistance surfaces, but no amphibian study analyzed forest cover, precipitation, roads or temperature independently; no bird study used habitat suitability models; no plant study assessed the effect of water bodies; no reptile or plant study addressed the effect of roads; and no mammal study considered temperature (Fig. 6).

Figure 6 Number of studies focusing on different landscape effects on gene flow for each taxonomic group.

See Dataset S1 for details on the reported effects.

The effect of habitat loss on gene flow was assessed in 25 studies and 39 species (Dataset S2). From these, only 14 studies detected a reduction of gene flow with increasing habitat loss in three plants, five mammals, one amphibian, two birds and one insect. Remarkably, most insects were unaffected by habitat loss. Only 11 articles reported microsatellite-based IBR effects, comprising 25 species (Dataset S3). Whereas IBD drove most significant effects across this group of studies, individual-level studies (N = 14 effects; Fig. 7) showed larger effect sizes than population-level ones (N = 78 effects; Fig. 8). Two individual-level IBR effects were significant (revealing isolation by elevation in a bird and an ant, Fig. 7), and three significant IBR effects were identified in population-level studies (revealing isolation by elevation in a plant and a bee, and isolation by non-forested areas in an ant, Fig. 8).

Figure 7 Individual-level effect sizes for isolation by geographic distance (A), isolation by elevation (B) and isolation by forest cover (C).

Dots represent effect sizes and colors indicate taxonomic groups. Significance of the effects reported in the original articles is also highlighted.

Figure 8 Population-level effect sizes for isolation by geographic distance (A), isolation by elevation (B), isolation by precipitation (C), isolation by temperature (D) and isolation by forest cover (E).

Dots represent effect sizes and colors indicate taxonomic groups. Significance of the effects reported in the original articles is also highlighted.

Discussion

Despite the extraordinary levels of biological diversity comprised in the tropics, the study of landscape effects on gene flow in tropical organisms only began to gain general attention in the past five years. Still, published studies are mainly concentrated in the Americas and most of them have focused on amphibians or mammals. The majority of studies were performed at the population-level, electrical resistance was the most common resistance metric employed, microsatellites were the most frequently employed type of molecular marker, and Mantel and partial Mantel tests the most common statistical approaches used. While elevation, land cover and forest cover were the most common gene flow predictors assessed, habitat suitability was found to be a common predictor of gene flow. A third of all surveyed studies explicitly assessed the effect of habitat degradation on gene flow, and only 14 studies detected a reduced gene flow with increasing habitat loss. Finally, individual-level microsatellite-based IBR effects showed higher effect sizes than population-level ones, elevation was responsible for most significant effects and a single study reported significant isolation by non-forested areas in an ant.

One of the main aims of the field of landscape genetics has been to understand how landscape characteristics shape patterns of functional connectivity (Manel & Holderegger, 2013), a subject that has been addressed by many studies undertaken in temperate regions (Balkenhol et al., 2016). Here we show that the study of landscape effects on gene flow in tropical organisms has lagged behind, and that published studies are concentrated in the Americas, as are general research effort on biodiversity in human-modified tropical forests (Gardner et al., 2009; Schlaepfer et al., 2018). Moreover, we found that amphibians and mammals were overrepresented in our surveyed studies, and most studies outside the Americas focused on mammals (Figs. 1 and 2), reflecting taxonomic biases in biodiversity data and societal preferences (Troudet et al., 2017). Our results thus highlight how little we still understand about landscape effects on gene flow in the tropics, and call for more studies on unrepresented taxonomic groups, tropical areas outside the Americas, and exclusively aquatic organisms.

Most of the surveyed studies used microsatellite markers, despite the not so recent shift towards genotyping by sequencing (GBS) triggered by next generation sequencing technologies (Allendorf, Hohenlohe & Luikart, 2010; Benestan et al., 2016). For instance, microsatellite genotyping is still cheaper than GBS, and cross-amplification of SSR markers in related species often reduces the cost of developing species-specific markers (Storfer et al., 2010). However, SNPs are rapidly becoming the new standard in population and landscape genomic studies, due to their genome-wide coverage and analytical simplicity (Morin, Luikart & Wayne, 2004). Moreover, sequencing costs have fallen dramatically (Shendure et al., 2017), and GBS approaches (such as RAD-sequencing) allow an affordable high-coverage sequencing of a representation of the genome and the discovery of thousands of SNPs in organisms lacking a reference genome (Rowe, Renaut & Guggisberg, 2011; Hohenlohe, Catchen & Cresko, 2012). Perhaps the most important obstacle preventing the widespread adoption of GBS is the complexity of bioinformatic processing (pre-processing of sequence data) and working with very large datasets (Johnson, 2009), but we believe that a much higher resolution coupled with the possibility to study both neutral and adaptive genetic variation are worth the effort (Rodriguez et al., 2015; Lanes et al., 2018).

Electrical resistance was the most common resistance metric employed, revealing its ample adoption as a general indicator of animal and plant gene flow (McRae & Beier, 2007). Additionally, we found that Mantel and partial Mantel tests were the most widely used statistical approaches to relate landscape and environmental characteristics with gene flow, even though better methods are available (Prunier et al., 2015; Richardson et al., 2016). The limitations of Mantel tests have been thoroughly discussed (Guillot & Rousset, 2013; Zeller et al., 2016), and include high type-I error rates (i.e., false positives), the inability to model the effect of multiple covariates simultaneously (as in a multiple regression), and the absence of a maximum-likelihood framework that allows for model selection (Shirk et al., 2010; Shirk, Landguth & Cushman, 2018). Maximum likelihood population effects (MLPE) are particularly appealing mixed-effects models for use landscape genetic studies because they allow implementing multiple regressions that account for the non-independence of pairwise distances within a likelihood framework (Clarke, Rothery & Raybould, 2002), compatible with model selection based on information criteria such as AIC (Jaffé et al., 2016; Row et al., 2017; Shirk, Landguth & Cushman, 2018).

Most surveyed studies assessed gene flow responses to few landscape and environmental variables, from which elevation, land cover and forest cover were the most common. For instance, no plant study assessed the effect of water bodies; no reptile or plant study addressed the effect of roads; and no mammal study considered temperature. Again, these finding suggest data and societal preferences (Troudet et al., 2017), although the more limited availability of environmental layers in tropical compared with temperate regions must be highlighted too. Making available more spatially explicit environmental data in the tropics could certainly help broaden the scope of future efforts to capture landscape effects on gene flow (Collen et al., 2008; Barlow et al., 2018). The surveyed studies were nevertheless able to quantify functional connectivity (Balkenhol et al., 2013; Carvalho et al., 2015; Ruiz-Lopez et al., 2015), propose ecological corridors (Atickem et al., 2013; Yumnam et al., 2014), assess threat status (Lanes et al., 2018), evaluate restoration effectiveness (Moraes et al., 2018), and forecast the impact of future climate and environmental changes on gene flow (Thomassen et al., 2009; Velo-Antõn et al., 2013; Thatte et al., 2018). Interestingly, several studies found an effect of habitat suitability on gene flow, suggesting that habitat suitability models are useful when proposing ecological corridors or forecasting the impact of future climate on gene flow (Franklin & Miller, 2009), although habitat suitability does not always reflect permeability for dispersal (Mateo-Sánchez et al., 2015). Additionally, elevation was responsible for most significant microsatellite-based IBR effects (Figs. 7 and 8), a result that suggests elevation is an important mediator of functional connectivity in tropical landscapes (Worboys, Francis & Lockwood, 2010).

Despite global concerns with the negative effects of habitat degradation on tropical biodiversity (Barlow et al., 2018), only 25 studies have so far explicitly assessed the effect of habitat degradation on gene flow. From these, only 14 found reduced gene flow with increasing habitat loss, and a single microsatellite-based study reported a significant isolation by non-forested areas in an army ant (Fig. 8). In contrast to other flying insects where both females and males disperse, army ant queens are permanently wingless, so gene flow is restricted and mainly driven by male dispersal (Jaffé, Moritz & Kraus, 2009; Pérez-Espona, McLeod & Franks, 2012). These findings suggest that the effect of habitat loss on gene flow is difficult to detect, as species with extremely restricted dispersal are more likely to show large effect sizes and thus be less susceptible to type-II errors (false negatives).

Many sources of variation could have influenced the detection of landscape effects on gene flow, including species-specific differences in dispersal ability and reproductive systems, historical processes underpinning genetic differentiation, different sample sizes, the resolution of the spatial data (grain size), the extent of the study area, sampling design, and time-lags in the responses to landscape changes (Anderson et al., 2010; Balkenhol et al., 2016; Schlaepfer et al., 2018). However, small sample sizes, limited information on the natural history of most studied species and inconsistencies in the way data was reported across studies preclude a quantitative assessment of the impact of these factors on our observed effect sizes (Dataset S3). Even though the majority of the surveyed studies employed population-level approaches, individual-level studies showed higher effect sizes, a finding that reinforces that individual-level analyses based on continuously distributed samples are more powerful and appropriate for landscape genetic studies (Landguth et al., 2010; Balkenhol et al., 2016). Additionally, studies that account for the underlying population structure or inter-population variations in effective population size (Ne) are more likely disentangle landscape from drift effects on gene flow (Prunier et al., 2017). This is because population-level metrics of genetic connectivity like the frequently used FST actually measure the balance between genetic drift on the one hand, and migration on the other. To the best of our knowledge, none of the analyzed studies accounted for variations in Ne between sample units when modeling IBR. This can be done by employing different distance metrics (such as conditional genetic distance (Dyer, Nason & Garrick, 2010)), by restricting IBR models to sample units belonging to the same genetic cluster (i.e., with the same Ne), by including a random effect specifying the nature of pairwise genetic distances (from sample units belonging to the same or different genetic clusters), or through gravity models that explicitly incorporate Ne or other node-level proxy of population size (DiLeo & Wagner, 2016; Zero et al., 2017).

Conclusions

Our study reveals important knowledge gaps regarding landscape effects on gene flow in tropical organisms, which prevent making cross-species generalizations. However, general patterns of genetic connectivity provide important insights into common barriers to gene flow or responses to land use changes (Poelchau & Hamrick, 2012; Wang, Glor & Losos, 2013; Paz et al., 2015; Jaffé et al., 2016; Lanes et al., 2018). Such knowledge is particularly important to inform conservation actions seeking to safeguard ecosystem function instead of a few target species (Manel & Holderegger, 2013). Our work nevertheless provides some useful guidelines to help fill these knowledge gaps: (1) Increased efforts are needed to study unrepresented taxonomic groups and tropical areas outside the Americas, as well as generate more spatially explicit environmental data in the tropics; (2) the adoption of genotyping by sequencing and individual-level approaches could substantially increase statistical power and shed light into both neutral and adaptive patterns of genetic variation; (3) using mixed-effects MLPE models to relate genetic and spatial data could minimize type-I errors, result in more accurate parameter estimates (which account for multiple landscape and environmental predictors), and help establish a common model-selection framework across landscape genetic studies (Row et al., 2017; Shirk, Landguth & Cushman, 2018); (4) explicitly modeling the impact of historical processes underpinning genetic differentiation, the resolution of the spatial data, and possible time-lags (DiLeo & Wagner, 2016; Waits, Cushman & Spear, 2016), could help provide more confidence in landscape effects on gene flow and make IBR estimates comparable across studies.

Supplemental Information

Dataset S1 Full set of compiled studies assessing landscape effects on gene flow in tropical organisms

Click here for additional data file.

Dataset S2 Subset of studies assessing the effect of habitat loss and fragmentation on gene flow

Click here for additional data file.

Dataset S3 Subset of microsatellite-based studies reporting isolation by resistance effects

See methods for details.

Click here for additional data file.

We thank Nathaniel S. Pope for statistical support, Flora Bittencourt, Gabriela Schmaedecke and two anonymous referees for improving earlier versions of this manuscript.

Additional Information and Declarations

Competing Interests

Author Contributions

Data Availability

Rodolfo Jaffé is an Academic Editor for PeerJ. The authors declare there are no other competing interests.

Waléria Pereira Monteiro performed the experiments, analyzed the data, prepared figures and/or tables, authored or reviewed drafts of the paper, approved the final draft.

Jamille Costa Veiga and Amanda Reis Silva performed the experiments, analyzed the data, prepared figures and/or tables, approved the final draft.

Carolina da Silva Carvalho and Éder Cristian Malta Lanes performed the experiments, analyzed the data, contributed reagents/materials/analysis tools, prepared figures and/or tables, approved the final draft.

Yessica Rico performed the experiments, contributed reagents/materials/analysis tools, approved the final draft.

Rodolfo Jaffé conceived and designed the experiments, analyzed the data, contributed reagents/materials/analysis tools, authored or reviewed drafts of the paper, approved the final draft.

The following information was supplied regarding data availability:

Raw data is available in the Supplemental Files.

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
