# Peer review of "Everything you always wanted to know about gene flow in tropical landscapes (but were afraid to ask)"

_PeerJ, doi:10.7717/peerj.6446_

## Round 0.1 · original submission · Major Revisions

Please, when you will revise the manuscript, pay particular attention to the remarks by rev#2 about the use of the term “gene flow” and to recommendations by both reviewers who ask for a general improvement of the discussion.

Reviewer 1 ·

Basic reporting

The study provides a comprehensive review of how landscape effect, particularly, anthropogenic disturbances influence the gene flow in tropical ecosystems. By analysing the studies across 157 tropical species, the authors conclude that the use of individual-based approaches rather than a community-level investigation is appropriate to understand the Landscape Effect on gene flow. Furthermore, the authors review the methodologies often used in the concerned field of investigation and suggest that next-generation sequencing and maximum likelihood-based analysis is appropriate to generate comparable results. In their conclusion, the authors highlight the lack of information for tropical species and emphasize that the future investigations must include tropical species across geography.

The topic of the investigation is definitely interesting, however, I feel that the paper does not introduce the subject adequately. As a result, the core question of the present investigation appears blurred. I have mentioned a few points below, which, I hope, would help to improve the manuscript and bring out the motivation for the investigation.

1. Authors need to expand on the Landscape Effect, with and without anthropogenic disturbances
2. Broadly discuss how the landscape effect influence gene flow using
3. Describe and link functional connectivity to the above discussions
4. Expand Isolation by Resistance, its mechanism and outcome in connection to the Landscape Effect
5. Optionally, you can introduce a flow chart describing the link across all that will be helpful in visualizing the importance of the present investigation

Experimental design

The methods for literature survey and analysis seem fine to me. However, I have concerns regarding the study design and in formulating the objectives of interest. Please see the comments in "Basic reporting"

Validity of the findings

The goals set out in the Introduction needs improvement for clarity
Similarly, bring a comprehensive outlook to the subject in the Discussion section

Moreover, the work, in its present form, seems incomplete and requires clarity on a conceptual platform to connect the ideas appropriately

Additional comments

The title seems quite uninformative. I suggest the authors modify the title to justify the objective of the study
Line 51 You may remove the 'comma' before ‘found’
Line 55-68 The phrases ‘habitat degradation’ and ‘environmental change’ have been used in association with the biodiversity response. Do these phrases convey the same message or they underlie different concepts. Add or modify the sentence for clarity
Line 76-77 In the earlier paragraph, authors have described community- and population-level studies for biodiversity response to habitat degradation. However, here, they have mentioned population and individual level studies. Please clarify or modify!

Reviewer 2 ·

Basic reporting

Overall, this manuscript deals with a very interesting and timely topic that is cross-disciplinary and definitely within the scope of the journal. This specific topic has also not been reviewed before. The intro nicely outlines the motivation for the study. Indeed, the entire manuscript is very well written.

Experimental design

I found no major flaws in the analysis, but I think more insights could be gained from the analyses.

Validity of the findings

I cannot recommend to accept the manuscript in its current form, because it is not correct to talk about “gene flow” when the reviewed studies actually analyze genetic structure, and also because I found the discussion to be too general and not specifically related to the findings of the shown analysis. I provide details comment on these issues under point 4 of this review form and encourage the authors to resubmit the manuscript after revisions.

Additional comments

Major comments
1) I do not like how the manuscript talks about “gene flow”, but actually compares measures of genetic differentiation and structure. Especially the population-based metrics (e.g., Jost D, FST etc.) are not only affected by gene flow, but also by drift, which is essentially a function of (effective) population size. This is well-known from basic population genetic theory, but the importance of this for landscape genetic inferences was recently highlighted by Prunier et al . - Methods in Ecology and Evolution “Contribution of spatial heterogeneity in effective population sizes to the variance in pairwise measures of genetic differentiation”). Similarly, individual-based metrics are not only influenced by gene flow, but also by mating schemes, space-use behavior etc., especially across small spatial scales. You need to explain this in the discussion of your manuscript, so that readers are not misinterpreting your results. I recommend to not talk about “gene flow” at all, but instead about “genetic connectivity” in a broader sense or even about “genetic structure”, and then explain what exactly this entails (i.e., gene flow and other processes).

2) One general concern I have is the combination of data/results from plant and animal species. I realize that it is currently difficult to draw cross-species conclusions, but I don’t think it is correct to combine results from flying and non-flying insects, from plants and animals, and from terrestrial and aquatic systems without at least discussing the importance of different species traits for landscape genetic patterns. Also, you mention biases in previous studies, but you also have a strong bias towards insects in your quantitative analysis (Fig. 7 and 8, Data S2 and S3). You need to discuss this in much more detail. There is an interesting paper by Lino et al. that was recently published in Mammalian Biology (“A meta-analysis of the effects of habitat loss and fragmentation on genetic diversity in mammals“). It focused in genetic diversity and mammals, but you still might want to check out that paper and potentially include it as one of your references.

3) In addition to considering species traits, some more details on the studies are warranted. For example, in lines 163-164: I think in order to make sense of such counter-intuitive results, readers would appreciate more details on a) the study design employed by each study, especially concerning the spatial scale of analysis and the sampling scheme, and b) the study species (see above). For example, a small scale analysis on flying insects is much less likely to find an effect of habitat loss than a large scale analysis of ground beetles. You explain this for one of the studies (on army ants) later in the discussion, but it is important for all the studies you use.

4) Finally, I found the discussion to be too general. It is basically a list of things that should happen in landscape genetics (use of improved methods, use of NGS approaches, considering time-lags…). The need to consider all of this has been highlighted before and it is not clear exactly how tropical systems are different or how your study added to this existing knowledge. Thinking some more about the implications of species traits and study design might help you to get more info out of the analyses, but I already think there is a bit more in your data than you report. Isn’t it interesting that the bias towards mammals is strongest outside of the Americas? Isn’t it surprising that most amphibian studies use haplotype data? You could also discuss that several studies have used habitat suitability to estimate resistances, but suitability is often a poor descriptor of functional connectivity, so you could advise against this for future tropical studies. I found it surprising that no study has looked at road effects in reptiles, and no amphibian study assessed dispersal via waterways. I also think that a more detailed overview of study aims and outcomes would be nice. You mention these briefly in lines 224-229, but it the info is missing from the results. Overall, I suggest to derive more novel insights from the results you have already obtained and can additionally obtain by considering species traits and study design more explicitly.


Minor comments:
57: In the sentence “While these metrics underpin ecosystem function, they are not always be…” – either delete the “be”, or change to “While these metrics underpin ecosystem function, they MAY not always be…”

119: Not sure why you are citing the Balkenhol et al. 2016 book here. Perhaps a better citation would be chapter 3 in that book (i.e., Waits & Storfer “Basics of population genetics: quantifying neutral and adaptive genetic variation for landscape genetic studies”. (Also, the Balkenhol et al book is listed twice in the reference list (a + b)…)

123: Fig. 1 does not show which studies are individual- vs. population-based.

139: Please explain why you did not perform a more formal meta-analysis.

145: But looking at Figure 2 and Data S1, it really seems like the bias is mostly towards amphibians, while mammals, reptiles and insects are all rather similarly represented. I’d change your conclusions accordingly. Also see my comment on Fig. 1 concerning a mammal bias outside of the Americas.

146-148: I don’t understand what this sentence means. Please reword to make it clearer.

165-166: It would be important to know whether studies tested for IBD separately, or after accounting for the effects of landscape variables. After all, straight-line and effective distances are often highly correlated and when testing for IBD separately, it will be supported, even if the actual driver is the landscape matrix.

170: Is “isolation by deforestation” the same as detecting a positive effect of forest cover on gene flow? If so, I would reword to “isolation by non-forest”, because non-forest is a pattern (just like elevation), while deforestation is a process.

176: Most studies have focused on amphibians.

182: Citing Dyer 2015 here is not appropriate, because he actually argued that thus far, landscape genetics has not evolved into a truly distinct and independent discipline. Aside, I suggest to change “interdisciplinary discipline“ to „interdisciplinary field“.

188: Amphibians yes, but mammals not too much more than insects or reptiles.

194-208: While I agree that the future will see many more landscape genetic studies using NGS approaches and resulting SNPs, this whole paragraph deals with a very general aspect of landscape genetic s and is only marginally related to your own study. I suggest to shorten this and move it to the very end of the discussion.

211: Might be good to mention some of these improved methods, especially some that were not yet mentioned by Richardson et al. (e.g., commonality analysis – Prunier et al.,Molecular Ecology (2015) 24, 263–283; extended causal modeling - Fourtune et al. Am Nat. (2018) 191:491-508). This is again something quite general.

213-214: This is incorrect. More than one variable can be partialled out in a partial Mantel test before assessing the effect of the focal variable. See e.g., the implementation of partial Mantel tests in r package ‘ecodist’.

215: But see Franckowiak et al. 2017 (PLoS ONE 12(4): e0175194) for a severe critique of model selection with pairwise data.

216: Add “models” after “population effects (MLPE)”

223: Again, I don’t think the Balkenhol et al. 2016 book is a good reference for this statement.

240: No, species with limited dispersal are not generally expected to be most impacted by habitat loss and fragmentation – it all depends on the scale of habitat loss, and species with high dispersal capability will also be highly susceptible to habitat loss. See also Evers and Didham (Biol. Rev. (2006), 81: 117–142 ; Fig. 2B) and Lino et al. (Mammalian Biology, A meta-analysis of the effects of habitat loss and fragmentation on genetic diversity in mammals).

263: I wonder whether it is desirable / realistic to have one modelling framework for all landscape genetic studies. I feel that landscape genetic studies are too diverse for this in terms of spatial and temporal sampling scheme, scale of sampling, landscape and species characteristics, etc.

Fig. 1: It looks as if the vast majority of studies outside of the Americas has focused on mammals. This is another bias you might want to report.

Fig. 3: Need to mention that 2018 was not yet complete by the time you did this analysis

Fig. 4: A pie chart would be more appropriate for this figure

Fig. 5: Need to explain the abbreviations of the different methods

Fig. 7: It is not clear to me whether the significances shown by the different symbols are for the effects size (as shown in S3), or for the statistical tests in the original studies.

Fig. 8: Not sure that the ordering of this is very informative. Would it be better to order by species names, or even to move this mix of taxa to the supplement and here focus on a few species for which multiple landscape variables were tested? In the text, you use Fig. 7 and 8 to compare effects sizes for individual- vs. population-based studies, but for that is would be better to show individual- and pop-based results on the same panel…

Data S1: The column “N samples” sometimes lists just a number, sometimes it states “individuals” and sometimes “specimen”. Please explain these differences. Also, it would be important to know what level of analysis the studies used (pops or individuals), how many analytical units they had (a study with 600 individuals conducting population-based analyses with 3 populations has an effective N of 3) and whether the study was included in your quantitative analysis (because the citations are not listed in S3…). Some of this info is presented in S3, but it would be easier for readers to have it all in one table. Finally, some of the studies have been conducted in aquatic environments, but this is never mentioned in the main text.

Data S2: Please explain how “deforestation” is different from “% forest cover”

Data S3: Need to explain the abbreviations for the genetic distance metrics. What does “Pop_ind” mean? The s.e. are not shown in the Figures, but I think they should be used to calculate confidence intervals.

---

## Round 0.2 · Minor Revisions

Thank you for submitting a revised version of your paper. I have still a couple of minor remarks to this version of your ms, that is generally improved.

Among others, the most relevant is the one posed by reviewer #2 when she/he says that "You must also incorporate a very clear statement where you explain that even though you are interested in gene flow, the studies you reviewed use various genetic metrics of genetic differentiation and structure".

Many thanks.

Reviewer 1 ·

Basic reporting

I am happy with the new version of the manuscript. I have gone through the authors’ response to the reviewers and happy with the modifications incorporated. I accept the manuscript in its present form with a few minor suggestions mentioned below, which can be incorporated during proof-reading.

P75-79: I suggest simplifying the sentence.
P89: Please check “specie’s”
P185: I suggest explaining the term “Electrical resistance” in landscape genetics which is used as an analogy to resistance in a network. This will bring clarification to the readers of the manuscript.

Experimental design

No comment

Validity of the findings

No comment

Additional comments

No comment

Reviewer 2 ·

Basic reporting

Basic reporting is fine.

Experimental design

Study design is fine.

Validity of the findings

See my general comments below.

Additional comments

The new version incorporates most of the previous reviewer comments, but not all. The manuscript generally has improved, however, I still think that one main issue has not adequately been addressed, i.e., the inappropriate use of the term “gene flow”. In your reply to that comment, you basically say that many previous studies have used the term incorrectly, so it’s okay to continue to do so. That is obviously a horrible way to argue. I also don’t see why the definition of landscape genetics by Balkenhol et al justifies your terminology. After all, that definition describes a broad field (in which one process of interest is gene flow), while you are conducting a certain analysis (involving various genetic metrics that are influenced by gene flow, among many other factors) from which you then try to make very specific inferences (concerning gene flow). I am fine with leaving the term in the title, but I still think you should be much more careful in your use of the term throughout the manuscript. You must also incorporate a very clear statement where you explain that even though you are interested in gene flow, the studies you reviewed use various genetic metrics of genetic differentiation and structure; that these are often used as surrogates of actual gene flow estimates; and that genetic differentiation/structure is also influenced by many other processes. This statement should be either in the beginning of the discussion, or in the methods. In your reply, you actually acknowledge that equating genetic structure with gene flow is not straightforward by stating “Genetic structure, on the other hand, may not necessarily comprise gene flow, as it can result from drift acting on isolated populations, adaptive processes or both” – so then why do you think it is okay to act in your manuscript like gene flow and measures of genetic structure are the same?

Additional /new comments:
“Interestingly, several studies found an effect of habitat suitability on gene flow, suggesting that habitat suitability models are useful when proposing ecological corridors or forecasting the impact of future climate on gene flow (Franklin & Miller, 2009)” – But you should state that this is generally not the case, as many species will still move through habitat that is not suitable to live in. See Scharf et al. (2018, doi: 10.1186/s40462-018-0136-2), Keeley et al (2017, doi: 10.1016/j.landurbplan.2017.01.007), Abrahms et al. (2016; doi: doi: 10.1111/1365-2664.12714), Mateo-Sanchez et al (2015; doi: 10.1007/s10980-015-0194-4), …

“…a limit of two predictor variables that can be simultaneously analyzed (in partial Mantel tests)…” – this is still incorrect. A partial Mantel test can be used to partial out as many covariates as you want, but you can only assess the impact of one variable that you do not partial out. Please correct this statement.

---

## Round 0.3 · accepted · Accept

In my opinion, the minor changes requested in the second revision round have been sufficiently addressed and your manuscript is now suitable for publication in PeerJ.

#